# Malnutrition, Cancer Stage and Gastrostomy Timing as Markers of Poor Outcomes in Gastrostomy-Fed Head and Neck Cancer Patients

**DOI:** 10.3390/nu15030662

**Published:** 2023-01-28

**Authors:** Diogo Sousa-Catita, Cláudia Ferreira-Santos, Paulo Mascarenhas, Cátia Oliveira, Raquel Madeira, Carla Adriana Santos, Carla André, Catarina Godinho, Luís Antunes, Jorge Fonseca

**Affiliations:** 1Grupo de Patologia Médica, Nutrição e Exercício Clínico (PaMNEC) do Centro de Investigação Interdisciplinar Egas Moniz (CiiEM), 2829-511 Almada, Portugal; 2GENE—Artificial Feeding Team, Gastroenterology Department, Hospital Garcia de Orta, 2805-267 Almada, Portugal; 3Residências Montepio—Serviços de Saúde, SA—Rua Julieta Ferrão Nº 10–5º, 1600-131 Lisboa, Portugal; 4Otorhinolaryngology Department, Hospital Garcia de Orta, 2805-267 Almada, Portugal

**Keywords:** head and neck cancer, nutritional status, percutaneous endoscopic gastrostomy

## Abstract

For percutaneous endoscopic gastrostomy (PEG)-fed head and neck cancer (HNC) patients, risk markers of poor outcomes may identify those needing more intensive support. This retrospective study aimed to evaluate markers of poor outcomes using TNM-defined stages, initial anthropometry [body mass index (BMI), mid-upper arm circumference (MUAC), tricipital skinfold (TSF), mid-arm muscle circumference (MAMC)] and laboratory data (albumin, transferrin, cholesterol), with 138 patients, 42–94 years old, enrolled. The patients had cancer, most frequently in the larynx (*n* = 52), predominantly stage IV (*n* = 109). Stage IVc presented a four times greater death risk than stage I (OR 3.998). Most patients presented low parameters: low BMI (*n* = 76), MUAC (*n* = 114), TSF (*n* = 58), MAMC (*n* = 81), albumin (*n* = 47), transferrin (*n* = 93), and cholesterol (*n* = 53). In stages I, III, IVa, and IVb, MAMC and PEG-timing were major survival determinants. Each MAMC unit increase resulted in 16% death risk decrease. Additional 10 PEG-feeding days resulted in 1% mortality decrease. Comparing IVa/IVb vs. IVc, albumin and transferrin presented significant differences (*p* = 0.042; *p* = 0.008). All parameters decreased as severity of stages increased. HNC patients were malnourished before PEG, with advanced cancer stages, and poor outcomes. Initial MAMC, reflecting lean tissue, significantly increases survival time, highlighting the importance of preserving muscle mass. PEG duration correlated positively with increased survival, lowering death risk by 1% for every additional 10 PEG-feeding days, signaling the need for early gastrostomy.

## 1. Introduction

Head and neck cancer (HNC) include cancers in the lips, mouth, nasal cavity, paranasal sinus, pharynx, larynx, and proximal esophagus, that share some common features. Most of them (90%) are squamous cell carcinomas related to tobacco smoking, heavy alcohol consumption, or human papillomavirus infections, and tend to affect swallowing and oral feeding. HNC patients present a very high risk of developing malnutrition for several reasons. First, tobacco smoking and heavy alcohol consumption are associated with malnutrition [1]. Moreover, heavy alcohol consumption frequently results in social disruption, which may lead to a further decline in nutritional status. HNC patients may be malnourished before cancer development due to these unhealthy habits.

The wasting effects of cancer have a major impact on nutritional status. Cancer malnutrition is considered as malnutrition associated with mild to moderate inflammation [2]. It is much more catabolic than simple starvation, with greater consumption of body lean mass and muscle proteins. HNC patients frequently present a reduced oral intake due to mechanical obstacles, causing dysphagia or odynophagia [3]. The mass position and cancer therapy may affect these patients’ chewing and swallowing. Therapeutic procedures like surgery, chemotherapy or radiotherapy could significantly impact nutritional status [3,4,5] by reducing food intake, contributing to malnutrition [6,7].

Malnutrition is very frequent (>70%) in HNC patients with severe weight loss and impaired immune function, leading to incomplete or postponed treatment cycles, and decreased quality of life [8]. Malnourished HNC patients present an increase in the number and severity of complications, and decreased survival [9,10]. Maintaining an optimal nutritional status is mandatory for improving treatment tolerance, outcome, and survival for all patients receiving cancer-directed treatment [11]. These patients suffering from malnutrition need specialized nutritional support. When oral intake is insufficient, and there is no other digestive tract disturbance, tube feeding is the obvious option. Most of these patients need it for some period during the evolution of the disease [12]. If tube feeding is required for more than 3 weeks, percutaneous endoscopic gastrostomy (PEG) is the gold standard. This is associated with fewer treatment failures and provides better nutritional support than long-standing nasogastric feeding tubes [13,14,15]. In HNC patients, PEG feeding improves clinical outcomes and survival [16].

HNC dysphagic patients frequently present speech deficiency that evolves parallelly with the swollen impairment. Speech difficulties may inhibit the use of several nutrition evaluation tools, and artificial feeding teams must often rely on objective data (e.g., anthropometric and laboratory data) for the nutritional status follow-up of PEG patients. Serum albumin, transferrin, and total cholesterol levels are non-specific, but may be used as serum markers of malnutrition, inflammation, and/or prognosis [17]. In fact, albumin and transferrin are negative acute-phase proteins and their production may be impaired in a long-term inflammatory stage, as well as in starvation. Although anthropometric and laboratory evaluation may reflect other influences that are diverse from nutritional issues, taken together they may become very useful tools for teams following PEG patients [18,19]. These tools are frequently used to assess the patient status, as they are low-cost, easy to obtain, and widely available [20,21,22,23,24].

Although the guidelines recommend an early gastrostomy of HNC patients [25], many of them present evident malnutrition when referred to the PEG procedure. We previously built a predictive model that helps us to identify patients with a probable life expectancy shorter than 3 weeks [18]. Additionally, we have previously identified nutritional and laboratory factors associated with poor outcomes for HNC patients after PEG [26]. These previous studies focused only on nutritional issues to identify prognostic factors and produce predictive models. In the present study, the classification of malignant tumors (TNM) was added to include the cancer severity and evolution. We remain interested in analyzing whether:

According to guidelines, is the patient a suitable candidate for gastrostomy, with a life expectancy longer than 3 weeks?How to use nutritional and laboratory data to identify HNC patients with severely impaired nutritional status, and unfavorable outcomes months after the gastrostomy, requiring more powerful nutritional support with larger protein energy intake?

In the present study, we aim to answer the question: can the cancer staging severity, anthropometric and laboratory data help us identify severely compromised patients requiring special attention?

Specifically, we aim to:

Evaluate the clinical and nutritional status of HNC patients when referred to endoscopic gastrostomy for long-term enteral nutrition, using anthropometry, laboratory data and accessible tools, even with patients who cannot speak.Evaluate the clinical outcome of PEG-fed HNC patients.Evaluate the relations between survival, severity, and nutritional status:Compare the nutritional status with the different TNM-defined stages.Compare the nutritional status with the different grades of stage IV, grouped as having metastases and no distant metastases (IVa and IVb, against IVc).Evaluate the impact of clinical and nutritional status on the survival of PEG-fed HNC patients, using TNM-defined stages, anthropometry, and laboratory data.

## 2. Materials and Methods

### 2.1. Patients

We studied consecutive adult HNC patients who underwent endoscopic gastrostomy to have PEG nutritional support, between January 2006 and December 2019 (the last year before the COVID-19 pandemic period). We included patients with cancers in the oral cavity, pharyngeal, laryngeal, esophageal proximal, and neck regions, arising from other organs or tissues. All patients were routinely evaluated in Garcia de Orta hospital Outpatient Artificial Nutrition Clinic, before the gastrostomy procedure, and one week, one month, and three months after the gastrostomy. After the third month, stable patients were followed every 4 to 6 months. Patients who experienced difficulties adjusting to PEG feeding were evaluated more often, until the patient and the caregiver achieved complete adaptation.

### 2.2. The Clinic, Anthropometric and Laboratory Data

All clinic, anthropometric and laboratory data are part of the routine evaluation of PEG patients, and were collected from the clinical files of the Artificial Feeding Team (GENE–Grupo de Estudo de Nutrição Entérica/parentérica). We recorded data on the day of the endoscopic gastrostomy or the day before. A blood sample was obtained in the endoscopy room, just before the gastrostomy procedure. Incomplete patient data was an exclusion criterion.

### 2.3. Head and Neck Cancer TNM Classification of Malignant Tumors

We searched data in electronic clinical files and the otorhinolaryngology oncology multidisciplinary reunion database for each patient by process number. Exclusion criteria were applied: no data for neoplasia location, incomplete cancer staging and advanced liver or kidney disorders.

Each patient’s cancer staging was obtained using the manual American Joint Committee on Cancer (AJCC) eighth edition to standardize the data. Each patient was classified within a TNM-defined stage: I, II, III, IVa, IVb, or IVc.

### 2.4. Clinical Outcome

According to the outcome, we divided patients into three categories: dead, lost to follow-up and alive. The time span from the gastrostomy procedure until death or until December 2019 was expressed in months.

### 2.5. Anthropometric Evaluation

The anthropometric evaluation was performed according to the International Society for the Advancement of Kinanthropometry manual on the day of the gastrostomy procedure or the day before. We obtained three consecutive measurements; the clinical file record represents the mean of those three measurements.

Body mass index (BMI) was obtained in most patients using the equation Weight (Kg)/Height (m)^2^. If patients were bedridden and could not stand up for weight and height evaluation, BMI was estimated using the mid-upper arm circumference (MUAC) and regression equations described by Powell-Tuck and Hennessy [27]; this method has been previously used and proved to provide a reliable BMI estimation in PEG patients [28,29]. Each patient was classified by the WHO classification according to their age as having low BMI if was <18.5 kg/m^2^ or < 22 kg/m^2^, normal BMI if 18.5–25 kg/m^2^ or 22–27 kg/m^2^, and high BMI if >25 kg/m^2^ or >27 kg/m^2^, for patients under 65 years or 65 years old or older, respectively [30] (Table 1).

**Table 1 nutrients-15-00662-t001:** Body mass index (BMI) classification according to age.

	Low	Normal	High
<65 Years	<18.5 kg/m^2^	≥18.5–<25 kg/m^2^	≥25 kg/m^2^
≥65 Years	<22 kg/m^2^	≥22–<27 kg/m^2^	≥27 kg/m^2^

2.Mid-upper arm circumference (MUAC) was evaluated using an inextensible measuring tape, with a 1 mm resolution. MUAC results were obtained from evaluating several tissues representing fat and lean mass.3.Tricipital skinfold (TSF), was measured using a Lange skinfold caliper with a 1 mm resolution. TSF evaluates the subcutaneous adipose tissue and estimates adipose reserves.4.The mid-arm muscle circumference (MAMC) was calculated according to the equation: MAMC = MUAC (cm) − 0.314 × TSF (mm). The MAMC allows us to estimate lean and muscle mass.

For each patient, MUAC, MAMC, and TSF were compared with reference values of the National Health and Nutrition Examination Survey (NHANES), through the comparison with the Frisancho reference tables [31,32,33,34].

### 2.6. Laboratory Evaluation

A blood sample was obtained from these patients, minutes before the endoscopic gastrostomy procedure. Blood samples were obtained between 8:00 and 10:00 A.M. following at least 12 h of fasting. Serum albumin <3.5 g/dL, serum transferrin <200 mg/dL, and total serum cholesterol <160 mg/dL were considered low values, suggestive of poor prognosis and/or malnutrition [18,35,36,37,38].

### 2.7. Statistics

All statistical analyses were computed by SPSS software version 26. Survival analysis (Kaplan Meier/Cox regression) provided all results evaluating the impact of covariates on PEG patient survival time. Linear regression analysis allowed us to estimate the impact of TMN-defined stages on HNC patient nutritional status biomarkers before PEG, by Z-testing the obtained marginal estimates. Statistical significance for each model and associated parameters were set at *p* < 0.05.

## 3. Results

### 3.1. Subjects

We enrolled 138 HNC patients (129 males, and 9 females), who underwent endoscopic gastrostomy to be PEG fed. Participants ages ranged from 42 to 94 years (mean: 61.3 years; median: 60.0 years). The characteristics of the study population, including the demographic data (age and gender), are presented in Table 2. HNCs arise from several organs: oral cavity (mouth), pharynx, larynx, and other organs and tissues. Patients presented HNCs at stages I to IVc.

### 3.2. Head or Neck Cancers

#### Cancer Location

For 129 males, the primary tumor was located in the pharynx (*n* = 41) or the larynx (*n* = 52). For the nine females, the primary tumor was mainly in the mouth (*n* = 5) (Table 3).

All patients were classified according to the TNM classification, from data searched in the clinical files of the otorhinolaryngology oncology multidisciplinary reunion, and an otorhinolaryngology specialist validated each datum (Table 3). The most frequent tumor stage was IV, present in 102 male and seven female patients.

### 3.3. Anthropometry

#### 3.3.1. Body Mass Index (BMI)

For eight patients, BMI was estimated using the Powell-Tuck and Hennessy regression equations. BMI ranged from 14 Kg/m^2^ to 48 Kg/m^2^ (mean: 20.64 Kg/m^2^; median: 19.4 Kg/m^2^). Classification was used according to age. Following this classification, 76 (55%) patients displayed a low BMI. The results are summarized in Table 3.

#### 3.3.2. Mid-Upper Arm Circumference (MUAC)

Compared with Frisancho criteria [31], 114 (83%) patients showed MUAC in the low range (Table 3).

#### 3.3.3. Tricipital Skinfold (TSF)

In this anthropometric parameter, 58 (42%) patients displayed low TSF (Table 3).

#### 3.3.4. Mid-Arm Muscle Circumference (MAMC)

In this anthropometric parameter, 81 (59%) patients showed MAMC in the low range (Table 3).

### 3.4. Laboratory Assessment

#### 3.4.1. Serum albumin

In 92 patients, albumin was in the normal range, and 47 displayed low serum albumin.

#### 3.4.2. Serum Transferrin

In 46 patients, transferrin was in the normal range, and 93 showed low serum transferrin.

#### 3.4.3. Total Serum Cholesterol

In 86 patients, cholesterol was in the normal range, while 53 patients displayed low serum total cholesterol. Of 130 males, 49 displayed a low serum total cholesterol. Of nine females, three displayed a low serum total cholesterol.

Laboratory data are summarized in Table 3.

### 3.5. Clinical Outcome

At the end of December 2019, out of the 138 patients, six (4.3%) were lost to follow-up, 111 (80.5%) patients were deceased, six (4.3%) were still PEG-fed and followed by the Artificial Nutrition Outpatients Clinic, and 15 (10.9%) resumed oral feeding with the tube removed and gastrostomy closed. Comparing all patients, the ones who had a longer survival time were patients with cancer classification TNM defined as stage I, and with the location of the cancer in the pharynx.

### 3.6. Kaplan–Meier Survival Analysis

Stage I cancer was associated with increased survival in any type of cancer. Stage III and IVa showed a similar survival time, and the stage with the least survivability was type IVc (Figure 1).

The mean survival time was 996 days (Table 4).

The pharynx appeared to be the type of cancer associated with the longest survival time, mainly in stages I and III. In stage IVc, any type of cancer had a much shorter life span (Table 5).

### 3.7. Cox Regression Analysis

We applied a Cox regression to obtain a statistical model adjusted for HNC TNM-defined stage to evaluate the tumor site, age, gender, anthropometrics, biochemical and PEG covariates effect on a patient’s survival time. Throughout the model fit process, the tumor site, age, and gender resulted in redundant variables and were removed from the reduced final model (Table 6).

Stage IVc was the only stage that had significance for the impact on survival time regardless of MAMC and PEG time (CI = [0.775, 0.901], *p* < 0.001; CI = [0.999, 0.999], *p* < 0.001). In the earlier stages (I, III, Iva and IVb), PEG time and MAM seemed to be major determinants of survival. Stage II was withdrawn due to having a single patient in this stage.

Patients with stage IVc had a four-times higher risk of death than those with stage I (OR 3.998).

The MAMC had an average odds ratio of 0.838, and each unit increase of MAMC was associated with a 16% decrease in the risk of death. The PEG duration time increasing by one unit was associated with a 0.1% reduction in the risk of death. That is, for every 10 days more of PEG (limited by the study time-frame), the risk of death decreased by 1%.

Globally, for all stages and cancer locations, the 3000-day survival rate was less than 10% (Figure 2).

### 3.8. Regression Analysis of Cancer Stage Impact on Nutrition Markers

We performed a linear regression analysis to estimate the impact of TMN-defined stages on HNC patient nutritional status biomarkers before PEG.

#### 3.8.1. TNM-Defined Stages (I vs. II vs. III vs. IVa vs. IVb vs. IVc)

Model results showed significant differences among the BMI (*p* = 0.039), and TSF (*p* = 0.007) of the TNM-defined stages. The BMI and TSF tended to decrease as the severity of the TNM-defined stages increased.

Model results showed no significant differences among the MUAC (*p* = 0.0231); MAMC, (*p* = 0.584); albumin (*p* = 0.165); transferrin (*p* = 0.074); and cholesterol (*p* = 0.035) of the TNM-defined stages. Nevertheless, MUAC, MAMC, albumin and transferrin tended to decrease as the severity of the TNM-defined stage increased. Cholesterol presented non-linear changes in the different TNM-defined stages.

#### 3.8.2. TNM-Defined Stages IVa and IVb vs. IVc

When comparing IVa, IVb vs. IVc, there were no significant differences in BMI (*p* = 0.169), MUAC (*p* = 0.149), MAMC (*p* = 0.307), TSF (*p* = 0.068) and cholesterol (*p* = 0.135).

Albumin and transferrin, when comparing IVa, IVb vs. IVc, showed significant differences (*p* = 0.042 and 0.008, respectively). Nevertheless, all parameters tended to decrease as the severity of the TNM-defined stage increased.

## 4. Discussion

Head and neck cancer (HNC) patients present a high frequency of malnutrition compared with other cancers, due to the direct effects of the disease, therapy side effects, and poor food intake [39]. HNC patients are often malnourished at diagnosis, having involuntary weight loss before starting treatment [40]. Good nutritional management is essential to the patient’s ability to complete the prescribed treatment courses, minimize nutrition-related side effects, and foster healing [41].

Our results show that this type of cancer affects most of the male gender, probably, due to poor lifestyle habits, such as smoking and alcohol consumption. Comparing the genders, both presented a high percentage of malnutrition. In the male gender, we can see in our study that a significant percentage of obese patients are likely linked to poor eating habits. In this study, we have a higher percentage of patients in advanced cancer stages (TNM defined as stage III and IV), characterized by the worst nutritional status and poor prognosis.

Although nutritional evaluation could benefit from sophisticated devices for measuring body composition, such as bioelectrical impedance analysis (BIA) or CT scan analysis, those devices were not available for all patients. Although less precise, BMI and anthropometry are inexpensive and widespread nutritional evaluation tools, classically used as an approach to the evaluation of fat/lean mass [33,34] and available everywhere, even in institutions with scarce resources. Most of our anthropometric data display low values, related to a poor nutrition status due to an advanced cancer stage. Arm anthropometry (MUAC, TSF, and MAMC) data show malnutrition in over eighty per cent of the patients. Estimation of fat and fat-free reserves, also reveals a poor nutritional status. MAMC recognizes more malnourished patients than TSF, which suggests that lean tissue is depleted at the beginning of the disease, fat reserves are more preserved, and over time, they are slowly degraded. Also, MAMC is an independent outcome predictor, highlighting the importance of lean mass in patient survival. From another perspective, MAMC is strikingly reduced since the early stages, and the other anthropometric and laboratory data reduce gradually, as the disease stage and severity progress [42].

Serum proteins are negative acute-phase proteins, and, like cholesterol, they may be modified by various biological influences. However, the usefulness of biochemical data is well recognized in several nutritional studies [17,20,26]. In our study, most patients display laboratory markers in a normal range, but albumin and transferrin tend to decrease with increasing severity of the TNM-defined stage.

Globally, our anthropometric and biochemical results demonstrate the strong influence that HNC had on the lean tissue and, later, on the fat mass of these patients, leading to malnutrition. Other authors have addressed the problem of malnutrition in cancer patients and their outcomes, such as the impact of nutrition management and status of head and neck cancer patients, on the success of treatment and survival [9,16,21].

Regarding the impact of the cancer stage (by TNM-defined stage) on the different nutritional parameters evaluated, when compared to all stages, only the BMI and the TSF had a significant difference between stages, with a progressive decrease as the severity of the cancer stage increases. In contrast, MUAC and MAMC are reduced since the early stages. This suggests that lean tissue is consumed during the initial stages of the disease, as expected in cancer-related inflammation. In contrast, fat tissue suffers a progressive loss, unlike fat-free mass, which is severely depleted since the beginning of cancer progression. When we focus on the most severe cancer stages (Iva, IVb and IVc), only albumin and transferrin had a significant difference, decreasing as the severity of the cancer stage increases. Likely, fasting is more severe in this advanced cancer stage than in less advanced stages [43].

When we tried to create a model that evaluates the role of all parameters of this study against the clinical outcome as survival time, only the MAMC was statistically significant, except for the most advanced cancer stage (Stage IVc). Therefore, this anthropometric parameter seems to no longer influence survival, as the severity of the disease increases to stage IVc. On the other hand, this result demonstrates the importance of preserving lean tissue in the early cancer stages, to maintain a better nutritional status and outcome. In fact, lean tissue is also associated with better treatment response and, consequently, a better prognosis [44,45,46]. Moreover, this study suggests that the PEG duration time positively impacts survival time in HNC patients. This supports the importance of early PEG feeding for HNC patient prognostics, suggesting that PEG feeding is important for better patient outcomes.

Early PEG is generally recommended in the treatment of HNC patients. Nevertheless, our results suggest that special attention should be addressed to patients with lower lean mass, evaluated through anthropometry, as in our study, or any other method.

Our study has some limitations resulting in some missing data. We completed processing patient data in December of 2019 because it was the last year before the COVID-19 pandemic, and several patients did not continue their follow-up, (refusing the hospital), and their records were incomplete. Other missing data included the causes of death. More than half of our patients (58.7%) were in TNM-defined stage IVa. The only TNM-defined stage II patient was excluded from inferential statistics to improve the statistical model parameter estimation.

## 5. Conclusions

HNC patients are malnourished when referred to undergo endoscopic gastrostomy and have advanced cancer as defined by TNM-defined stage, a marker of poor outcomes.

MAMC, the anthropometric parameter reflecting the lean tissue, was the only one with statistical significance in survival time, highlighting the necessity to preserve the muscle mass of these patients. PEG duration time was shown to correlate with increased survival time, at a rate of 1% decrease in the risk of death for every 10 days of PEG extension, suggesting that gastrostomy should be performed in an early stage of the disease progression.

## Figures and Tables

**Figure 1 nutrients-15-00662-f001:**
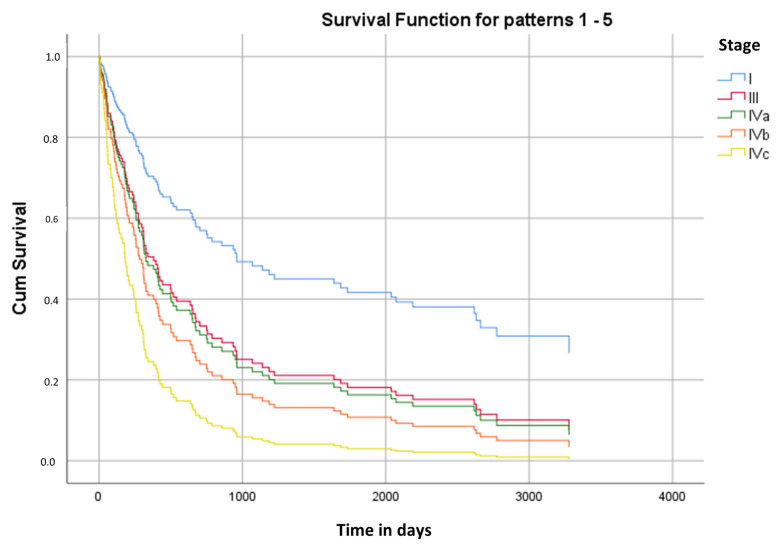
Kaplan–Meier curve of the cumulative survival for different cancer stages.

**Figure 2 nutrients-15-00662-f002:**
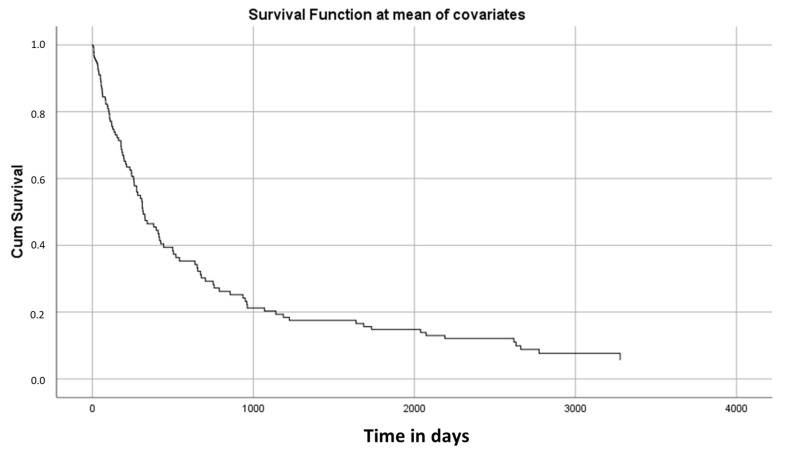
Kaplan–Meier survival curve in all stages and cancer location.

**Table 2 nutrients-15-00662-t002:** Subject characteristics.

Subject Characteristics	*n* (%)
Age (years)	
42–94 (mean 61.3)	
Gender	
Female	9 (6.5%)
Male	129 (93.5%)
Cancer site	
Mouth	39 (28.3%)
Pharyngeal	42 (30.4%)
Laryngeal	55 (39.9%)
Others	2 (1.4%)

**Table 3 nutrients-15-00662-t003:** Characterization of subjects by primary tumor location, TNM classification, anthropometry and laboratory serum data.

	Total (*n* = 138)	Total Mean
Primary Tumor located		
Mouth	39	
Pharynx	42	
Larynx	55	
Other HNC location	2	
Classification of Malignant Tumors (TNM) Classification		
Stage I	6	
Stage II	1	
Stage III	22	
Stage Iva	81	
Stage IVb	12	
Stage IVc	16	
Anthropometry Results		
BMI	76 Low BMI	
	46 Normal BMI	
	16 High BMI	
MUAC	114 Low	
	24 Normal	
TSF	58 Low	
	80 Normal	
MAMC	81 Low	
	57 Normal	
Laboratory serum data		
Albumin	47 Low	3.7 g/dL
	91 Normal	
Transferrin	93 Low	182.0 mg/dL
	45 Normal	
Total Cholesterol	53 Low	173.2 mg/dL
	85 Normal	

(BMI)—Body mass index; BMI classification according to age, <65 y, low BMI is <18.5 Kg/m^2^, normal BMI is between 18.5 Kg/m^2^ and <25 Kg/m^2^, and high BMI is ≥25 Kg/m^2^, ≥65 y, low BMI is <22 Kg/m^2^, a normal BMI is between 22 Kg/m^2^ and <27 Kg/m^2^, and high BMI is ≥ 27 Kg/m^2^; (MUAC)—mid-upper arm circumference <90% low, ≥90–110% normal; (TSF)—tricipital skinfold results, <90% low, ≥90–110% normal and (MAMC)—mid-arm muscle circumference <90% low, ≥90–110% normal; albumin < 3.5 g/dL (low), transferrin < 200 mg/dL (low), total cholesterol < 160 mg/dL (low).

**Table 4 nutrients-15-00662-t004:** Means and Medians for survival time (days) by stage of cancer.

Stage (N)	Mean				Median	
	Survival Time	Std. Error	95% Confidence Interval		Survival Time	Std. Error
			Lower Bound	Upper Bound		
I (6)	1135	496	164	2106	178	515
III (22)	1074	291	503	1645	275	268
IVa (81)	1054	149	762	1347	397	93
IVb (12)	1082	431	237	1927	233	202
IVc (16)	219	55	111	327	135	26
Overall (137)	996	117	767	1226	316	56

**Table 5 nutrients-15-00662-t005:** Median survival time by cancer location and stage.

				Time in Days
**Local**	Mouth	Stage	I (*n* = 1)	555
			III (*n* = 6)	75
			IVa (*n* = 27)	450
			IVb (*n* = 2)	400
			IVc (*n* = 3)	175
	Pharynx	Stage	I (*n* = 1)	2700
			III (*n* = 4)	2000
			IVa (*n* = 25)	305
			IVb (*n* = 6)	1700
			IVc (*n* = 5)	120
	Larynx	Stage	I (*n* = 3)	950
			III (*n* = 11)	631
			IVa (*n* = 29)	452
			IVb (*n* = 4)	250
			IVc (*n* = 8)	200
	Other locations	Stage	I (*n* = 1)	150
			III (*n* = 1)	150

**Table 6 nutrients-15-00662-t006:** Cox regression analysis.

		Coef	SE	*p*-Value	OR	95.0% CI for OR	
						Lower	Upper
TNM Stage	I			0.117			
	III	0.668	0.583	0.252	1.949	0.622	3.276
	IVa	0.728	0.529	0.169	2.071	0.734	3.408
	IVb	0.934	0.616	0.130	2.546	0.761	4.331
	IVc	1.386	0.592	0.019	3.998	1.252	6.744
MAMC		−0.177	0.040	0.000	0.838	0.775	0.901
Albumin		0.251	0.162	0.122	1.285	0.935	1.635
Time with PEG		−0.001	0.000	0.000	0.999	0.999	0.999

Mid-Arm Muscle Circumference (MAMC); percutaneous endoscopic gastrostomy (PEG); Coef variable or level coefficient; standard error (SE), Ref reference level; Odds ratio (OR).

## Data Availability

The data presented in this study are available on request from the first author.

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
