# Peer review of "Malnutrition, Cancer Stage and Gastrostomy Timing as Markers of Poor Outcomes in Gastrostomy-Fed Head and Neck Cancer Patients"

_nutrients, 2023, doi:10.3390/nu15030662_

Round 1

Reviewer 1 Report

Thank you for the interesting study. In order to improve readability and stress the focus of your article, I propose a thorough shortening of all sections and a reduction of table number by summarising severla results within one table. Due to long text and multiple tables, the reader is in danger of loosing the focus of your article. 

For the introduction:

Please shorten the introduction and reduce additional information, that ist not absolutely necessary to understand the results of your articel. In particular the paragraphe on the TNM stanging could be assuemd to be known or presented in a table. 

The list of aims is not very reader friendly. I would recommend to describe less aims, yet stress your most im portant research question. Please omit the reference to an extention of survival time (line 114) as your study does not have the power to answer this question.

Materials and methods

lines 195-200: Should be integrated whitin the discussion rather the method's section

Results: Due to the abundance of different results, the main research question does not get not pointed out. 

I wold recommend to summarise table 2-6 in a table displaying classic baseline characteristics of the population. The differentiation of male and female patients could be omitted to to very low numbers of female patients in total and its consecutive risk of bias in all results. 

Table 7. Why do you presnet mean and median?

Table 8. Please reevaluate the importance of Table 8. The interpretation is limited by missing information of numbers of patients in each categroy. Could the table be relocated within supplementary material?

Table 9: Should be held out to be the center of the analysis. Please add confidence interval and p-value to the results mentioned within the prosaic description of the results. Reduce prosaic description to the most important points of your central research question. 

Figure 2. Reevaluate the necessity of its presentation within the main results. Please rephrase title and description in more detail. 

Discussion: The discussion is very similar to the introduction, yet ist missing actual discussion of your results in the context of current literature. Please add a paragraphe with a summary of your key results and reduce the discussion to the most important points around the central research question. 

How do you interpret the increase of survival time in patients with PEG?

According to your data. Which parameters in addition to tumor stage do you recommend to consider in order to decide whether to start PEG? 

references: Ref. 17 (line 490-492) is rahter old. Is there a more recent publication to the still highly disputed importance of albumine as an marker of malnutrition?

Author Response

We appreciate all suggestions to the reviewer.

Thank you for the interesting study. In order to improve readability and stress the focus of your article, I propose a thorough shortening of all sections and a reduction of table number by summarising severla results within one table. Due to long text and multiple tables, the reader is in danger of loosing the focus of your article. 

  1. For the introduction:

Please shorten the introduction and reduce additional information, that ist not absolutely necessary to understand the results of your articel. In particular the paragraphe on the TNM stanging could be assuemd to be known or presented in a table. 

Answer: We appreciate feedback and have improved (lines 38-107).

The list of aims is not very reader friendly. I would recommend to describe less aims, yet stress your most important research question. Please omit the reference to an extention of survival time (line 114) as your study does not have the power to answer this question.

Answer: We appreciate comment and have improved.(line 97 to 107)

  1. Materials and methods

lines 195-200: Should be integrated whitin the discussion rather the method's section

Answer: We thank the alert and have corrected. It is now in discussion section (line342-347).

  1. Results: Due to the abundance of different results, the main research question does not get not pointed out. 

I wold recommend to summarise table 2-6 in a table displaying classic baseline characteristics of the population. The differentiation of male and female patients could be omitted to to very low numbers of female patients in total and its consecutive risk of bias in all results. 

Answer: We appreciate the feedback and have improved omitting the differentiation between genders.  (table 3 line 241).

Table 7. Why do you presnet mean and median?

Answer: table 7 is now table number 4 – Because both are informative, have different meanings, and are therefore reported.

The mean survival time is estimated as the area under the survival curve in the interval 0 to tmax (Klein & Moeschberger, 2003). Is a statement about the observed times. It shouldn't be taken to mean the length of time a subject can be expected to survive.

The median survival is the smallest time at which the survival probability drops to 0.5 (50%) or below. The median survival time and its 95% CI is calculated according to Brookmeyer & Crowley, 1982. Measuring the median survival is one way to see how well an intervention works.

Table 8. Please reevaluate the importance of Table 8. The interpretation is limited by missing information of numbers of patients in each categroy. Could the table be relocated within supplementary material?

Answer: We appreciate but we believe it's important to keep this table since it demonstrates the significant differences between the different cancer sites and which ones have the best prognosis.(Now is table 5)

Table 9: Should be held out to be the center of the analysis. Please add confidence interval and p-value to the results mentioned within the prosaic description of the results. Reduce prosaic description to the most important points of your central research question. 

We appreciate comment and agree. We have improved, please see table 6 and lines 284-287.

Figure 2. Reevaluate the necessity of its presentation within the main results. Please rephrase title and description in more detail. 

We have improved (line 309).

  1. Discussion: The discussion is very similar to the introduction, yet ist missing actual discussion of your results in the context of current literature. Please add a paragraphe with a summary of your key results and reduce the discussion to the most important points around the central research question. 

We appreciate the comment and have improved the discussion section (lines 328-398).

How do you interpret the increase of survival time in patients with PEG?

Thank you for the comment. We have improved this interpretation in lines 386-388

According to your data. Which parameters in addition to tumor stage do you recommend to consider in order to decide whether to start PEG? 

Thank you for the comment. We had this explanation in lines 389-391.

references: Ref. 17 (line 490-492) is rahter old. Is there a more recent publication to the still highly disputed importance of albumine as an marker of malnutrition?

We appreciate and improve replacing it with a newer one

Reviewer 2 Report

The original research conducted presents the impact of gastrostomy in patients with head and neck cancer and whether it may impact survival and quality of life.

Although the research provides some new insights, it is not novel per se as the current guidelines for placing PEG in HNC patients is well established in regards to earlier use prolonging survival in those with lower stages.

Throughout, the authors have to improve the English diction and sentence structure. It would benefit the phrasing to use a native English speaker or professional service.

The title spells "gastrostomy" wrong - please correct.

In the abstract, any used abbreviations have to be spelled out - several parameters that were measured are currently not. The phrase "death risk" should be replaced with mortality.

Page 2, line 55: the phrase "much more catabolic than simple starvation". should be rephrased and made more specific - "much more catabolic" should be further explained.

Page 2, lines 79-80: it is unclear which of the general measurements (albumin, transferring, or cholesterol) is actually an indicator of inflammation. Albumin by itself as a protein does not serve that purpose. Generally speaking, inflammation is characterized by C-reactive protein or cytokines. These do not appear to have been measured in this study.

Page 3, line 93: "more powerful nutritional support." Unclear what more powerful nutritional support constitutes. Please rephrase to be more specific in regards to what constitutes more powerful.

Page 4, line 174: please make sure to put all square as superscripts, such as height2. Also, please state how weight and height were measured (in kg or lbs. and cm or inches).

Page 5, line 190: the calculation for MAMC is unclear - please clarify.

Page 7, table 5: please include somewhere in the table the definitions for low, normal, and high for each parameter.

Page 8, figure 1 and page 11, figure 2: please submit both figures in higher resolution - the legends and text are not clearly readable.

Page 9, table 7: what does the superscript "a" stand for? Please define in the legends.

Page 10, table 9: the OR does not appear to match with the confidence intervals in terms of their significance. How can the OR for time with PEG be 0.999 with a CI of 0.999 and be significant? Please explain or correct.

It would benefit the paper if the discussion would include more current literature to refer to similar papers that have been published on this topic.

Author Response

Answers to the Reviewer 2

The original research conducted presents the impact of gastrostomy in patients with head and neck cancer and whether it may impact survival and quality of life.

Although the research provides some new insights, it is not novel per se as the current guidelines for placing PEG in HNC patients is well established in regards to earlier use prolonging survival in those with lower stages.

Throughout, the authors have to improve the English diction and sentence structure. It would benefit the phrasing to use a native English speaker or professional service.

The title spells "gastrostomy" wrong - please correct.

Answer: Sorry it was a typo, we have now the corrected word.

In the abstract, any used abbreviations have to be spelled out - several parameters that were measured are currently not. The phrase "death risk" should be replaced with mortality.

Answer: We appreciate the comment and have improved (line 27). We spelled out all the abbreviations in abstract.

Page 2, line 55: the phrase "much more catabolic than simple starvation". should be rephrased and made more specific - "much more catabolic" should be further explained.

Answer: Thank you, we agree and have improved (lines 48-50).

Page 2, lines 79-80: it is unclear which of the general measurements (albumin, transferring, or cholesterol) is actually an indicator of inflammation. Albumin by itself as a protein does not serve that purpose. Generally speaking, inflammation is characterized by C-reactive protein or cytokines. These do not appear to have been measured in this study.

Answer: We appreciate the comment and have improved (lines 73-75).

Page 3, line 93: "more powerful nutritional support." Unclear what more powerful nutritional support constitutes. Please rephrase to be more specific in regards to what constitutes more powerful.

Answer: We appreciate and improved. (line 92)

Page 4, line 174: please make sure to put all square as superscripts, such as height2. Also, please state how weight and height were measured (in kg or lbs. and cm or inches).

Answer: Sorry it was a formatting error. We have now corrected.

Page 5, line 190: the calculation for MAMC is unclear - please clarify.

Answer: We thank the comment and have correct. It was also a formatting error.

Page 7, table 5: please include somewhere in the table the definitions for low, normal, and high for each parameter.

Answer: We appreciate the comment and have improved. The table 5 is now the table 3.

Page 8, figure 1 and page 11, figure 2: please submit both figures in higher resolution - the legends and text are not clearly readable.

Answer: We appreciate the feedback and have improved the images definition.

Page 9, table 7: what does the superscript "a" stand for? Please define in the legends.

Answer: Sorry it was a formatting error.

Page 10, table 9: the OR does not appear to match with the confidence intervals in terms of their significance. How can the OR for time with PEG be 0.999 with a CI of 0.999 and be significant? Please explain or correct.

Answer: We appreciate comment and have improved. Now CI are fully reported to explicit the significance.

It would benefit the paper if the discussion would include more current literature to refer to similar papers that have been published on this topic.

Answer: We appreciate the feedback and have improved the discussion section (lines 328-398).

Round 2

Reviewer 1 Report

Thank you for the careful revision and the improvement of the presentation of your results in particular.